# Reliability of Free Inflation and Dynamic Mechanics Tests on the Prediction of the Behavior of the Polymethylsilsesquioxane–High-Density Polyethylene Nanocomposite for Thermoforming Applications

**DOI:** 10.3390/polym12112753

**Published:** 2020-11-21

**Authors:** Fouad Erchiqui, Khaled Zaafrane, Abdessamad Baatti, Hamid Kaddami, Abdellatif Imad

**Affiliations:** 1École de Génie, Université du Québec en Abitibi-Témiscamingue, 445, boul. de l’Université, Rouyn-Noranda, QC J9X 5E4, Canada; Khaled.Zaafrane@uqat.ca (K.Z.); Abdessamad.Baatti@uqat.ca (A.B.); 2Faculté des Sciences et Techniques, Université Caddi Ayad, Marrakech 40000, Morocco; h.kaddami@uca.ma; 3Unité de Mécanique de Lille, Université de Lille, UML, Joseph Boussinesq, F5900 Lille, France; abdellatif.imad@polytech-lille.fr

**Keywords:** thermoforming, PMSQ–HDPE, viscoelastic, experimental, bubble inflation test, DMA, Christensen’s model, FEM

## Abstract

Numerical modeling of the thermoforming process of polymeric sheets requires precise knowledge of the viscoelastic behavior under conjugate effect pressure and temperature. Using two different experiments, bubble inflation and dynamic mechanical testing on a high-density polyethylene (HDPE) nanocomposite reinforced with polymethylsilsesquioxane HDPE (PMSQ–HDPE) nanoparticles, material constants for Christensen’s model were determined by the least squares optimization. The viscoelastic identification relative to the inflation test seemed to be the most appropriate for the numerical study of thermoforming of a thin PMSQ–HDPE part. For this purpose, the finite element method was considered.

## 1. Introduction

The forming of thermoplastics in the plastic processing industry generally requires a high number of experimental tests to detect optimal conditions for mass production of products or optimizing of the manufacturing process. These experimental tests are costly and time-consuming. To circumvent the costs associated with these tests, many manufacturers are deploying computer-assisted analysis for product design [1]. However, computer-assisted analysis of the processing of polymers and composites demonstrates the need for an accurate description of the behavior of these materials under the combined effect of applied forces and temperature [2]. The quality of behavioral characterization depends largely on the tools used in experimentation, modeling, and optimization. Regarding the behavior of thermoplastics used in thermoforming, associated with the manufacture of thin parts, it is generally of a viscoelastic nature and the generated strains can be linear or nonlinear [3]. Several behavioral laws are available in the scientific literature to represent thermoplastic polymers. Among them are Maxwell [4], Christensen [5], K-BKZ [6], and Lodge [7]. These laws are generally constructed by combining the elastic and viscous responses of thermoplastics, in terms of spring and damper-based models.

For the numerical characterization of the viscoelastic behavior of materials, experimental data from rheological and mechanical tests are often used [8]. Concerning the experimental tests used for viscoelastic identification, there are two classes in particular: unidirectional tests [9,10,11] (dynamic mechanical tests in shear and extension, compression, etc.) and multidirectional tests [12,13,14,15,16] (inflation of circular and cylindrical membranes, equibiaxial stretching of membranes, extensions and simultaneous inflations of membranes, etc.).

At the level of the numerical identification of non-linear mechanical parameters, associated with the laws of viscoelastic behavior of thermoplastics, it is often necessary, with the help of mathematical modeling (analytical or numerical) and optimization, to reproduce, as faithfully as possible, the data measured in experimentation. Among the methods used for numerical modeling, the finite element method [2,11] and finite difference method [2] are used to model the experimental tests. Concerning the problem of identifying mechanical parameters by optimization algorithms, two classes are encountered: the class based on least squares algorithms [2,15,16] and the approach using artificial intelligence (neural networks) [12].

The deformations induced in thermoplastics, in the thermoforming process, are significant and, in general, of a biaxial nature. However, several works encountered in the literature on the construction of viscoelastic constitutive laws are based on experimental data from dynamic mechanical test (DMA). Thus, the following question arises: are the rheological data resulting from DMA tests reliable for the construction of a viscoelastic law? It is in this context that the present work is oriented and aims at a study on the reliability of the results obtained from two experimental tests: one was based on the inflation of the membrane and the other on a dynamic mechanical test (DMA). The two experimental tests were carried out at a temperature of 130 °C. For the viscoelastic characterization, we considered the Christensen model [5]. The mechanical parameters were identified using the Levenberg–Marquardt algorithm [17].

For the comparative study of the reliability of the results of the viscoelastic identification, compared to each experimental test, we considered the numerical modeling of the thermoforming of a thin part in PMSQ–HDPE. For this purpose, the finite element method was considered.

## 2. Material

This work is part of the work carried out on the development of a nanocomposite family based on polymethylsilsesquioxane (PMSQ, synthetized in our previous work [18]) and a high-density polyethylene (HDPE Hival-500354 with a melt flow index of 0.03 g min-1 (ASTM D1505) and a density of 0.954 g cm-3 (ASTM D1238) was supplied by IDES Prospector North America) matrix [19]. The method for the development of HDPE–PMSQ nanocomposites is based on a fusion mixing process. To this end, PMSQ nanoparticles were swollen in an organic solvent using an UltraTurax system (IKA, Wilmington, NC, USA) and sonication, then mixed with molten HDPE using a twin-screw extruder (Coperion corporation, Sewell, NJ, USA). Then, the solvent was removed. Nanocomposites with different PMSQ contents (from 0 to 1%) were manufactured. Then, the nanocomposites were characterized (Fourier transformation in the infrared, Perkin Elmer, Woodbridge, ON, Canada; transmission electron microscopy, JEOL, Tokyo, Japan; differential scanning calorimetry, Mettler Toledo, Greifensee, Suisse; scanning electron microscopy, JEOL, Tokyo, Japan; mechanical tests, TA Instruments, New Castle, DE, USA; thermophysical characterization, TA Instruments, New Castle, DE, USA). The mechanical properties obtained from HDPE–PMSQ nanocomposites were compared with the barrier effect of PMSQ nanoparticles. The elastic modulus, yield stress, and elongation at break of the neat HDPE and its nanocomposites are shown in Table 1. Compared to HDPE, the modulus of elasticity of HDPE–PMSQ was slightly improved.

In this present work, only the rheological properties obtained for a concentration of 1% of PMSQ were considered to detect the viscoelastic behavior of HDPE–PMSQ nanocomposites in the semi-solid state.

## 3. Experimental Testing

### 3.1. Bubble Inflation Testing

For the free blowing test, we considered a circular PMSQ–HDPE composite membrane. The diameter and thickness of the membrane were 80 and 1.5 mm, respectively (Figure 1). The description of the set-up and the experimental test procedures are described in [20]. Figure 2, extracted from [20], shows the experimental set-up diagram. Figure 3 shows the experimental results of the evolution, over time, of the internal pressure and the height at the pole, respectively, of the PMSQ–HDPE membrane.

### 3.2. Dynamic Mechanical Testing

In our study, the oscillatory shear experiment was performed to determine the elasticity or storage modulus (G’) and the loss modulus (G”) of the PMSQ–HDPE material. The results obtained with respect to the frequencies are given in Figure 4 at a temperature of 130 °C.

## 4. Viscoelastic Behavior Model

In this work, Christensen’s model [5], suitable for representing the viscoelastic behavior of thermoplastics in the semi-solid state, was used. For this model, in Lagrangian formulation, the second Piola–Kirchhoff stress tensor **S** at time *t* is given by:(1)$$S\left(t\right)=-p\left(t\right)C^{-1}+g_{0}I+\int_{-\infty }^{t}g_{1}\left(t-\tau \right)\frac{\partial E\left(t-\tau \right)}{\partial \tau }d\tau$$

**E** is the Lagrangian strain tensor **E**. *g*_0_ is the hyperelastic modulus and $g_{1}$ is the material relaxation function given by equation:(2)$$g_{1}\left(t-\tau \right)=\underset{k}{\sum }C_{k}e^{-\frac{t-\tau }{\tau_{k}}}$$
where *C_k_* is the stiffness modulus. The Lagrangian strain history **E** is related to Cauchy tensor deformation **C** by **E** = 1/2(**C**−**I**) and **I** is identity tensor.

The tensor **S** is related to the Cauchy stress tensor **σ** by the following relationship:(3)$$S\left(t\right)=J\left(t\right)F^{-1}\left(t\right)\sigma \left(t\right)F^{-T}\left(t\right)$$

**J**(t) and **F**(t) are, respectively, the Jacobian of the transformation and deformation gradient tensor. For incompressible materials, det(**J**(t)) = 1. For the blowing modeling of the PMSQ–HDPE membrane, we considered the following two assumptions:The state of plane stress;Material is incompressible.

The first hypothesis induces the following forms for the matrices E(t) and S(t):(4)$$C\left(t\right)=\left[\begin{array}{ccc} C_{xx}\left(t\right) & C_{xy}\left(t\right) & 0\\ C_{yx}\left(t\right) & C_{yy}\left(t\right) & 0\\ 0 & 0 & C_{zz}\left(t\right) \end{array}\right];\;S\left(t\right)=\left[\begin{array}{ccc} S_{xx}\left(t\right) & S_{xy}\left(t\right) & 0\\ S_{yx}\left(t\right) & S_{yy}\left(t\right) & 0\\ 0 & 0 & S_{zz}\left(t\right) \end{array}\right]$$

With the assumption of incompressibility of the PMSQ–HDPE composite, the term *C_zz_*(*t*), appearing in Equation (4), can be directly calculated from the other components of the strain tensor **C**:(5)$$C_{zz}\left(t\right)=\lambda_{3}^{2}\left(t\right)=\frac{1}{C_{xx}\left(t\right)C_{yy}\left(t\right)-C_{xy}\left(t\right)C_{yx}\left(t\right)}$$

λ_3_ is the principal stretch ratio in thickness direction defined by:(6)$$\lambda_{3}\left(t\right)=\frac{h\left(t\right)}{h_{0}}$$
where *h*(*t*) and *h*_0_ represent the PMSQ–HDPE membrane thicknesses in the deformed and undeformed configurations, respectively.

## 5. Viscoelastic Model Identification

### 5.1. PMSQ–HDPE Viscoelastic Behavior Identification Conforms to Bubble Testing

The mathematical formulation of the problem is described in [2,12]. To this end, the deformation of the circular membrane was assumed to remain axisymmetric during inflation. The strategy used for identification was as follows: first, for a given experimental thickness, the theoretical blowing pressure of the membrane, compatible with the measured thickness, was determined. For this purpose, we used the finite difference method with variable pitch. Then, using a modified Levenberg–Marquardt algorithm [17], the difference between the calculated and measured inflation pressure was minimized. Using this procedure, the material constants C_0_, g_b_, and τ_b_ were determined. However, it is important to note that the resolution of the equilibrium equations, which govern membrane inflation, can induce instabilities [12] that affect the numerical resolution. The choice of the initial values of the material constants is crucial for the convergence of the problem. As the experiment was based on a single average air flow rate for blowing the membrane, we considered a single relaxation time. In other words, three parameters for Christensen’s model were determined: C_0_, C_1_, and τ_1_.

The pressures and heights measured from the bubble to the pole were interpolated by polynomial functions and used in the identification problem. The predictions obtained with Christensen’s model gave very satisfactory results and are presented in relation to the experimental data in Figure 5. The mechanical properties obtained by numerical identification are given in Table 2.

Figure 6 illustrates, according to Christensen’s viscoelastic model, the main geometrical results relative to the PMSQ–HDPE membrane trace, at 0.05, 0.10, 0.15, 0.20, and 0.25 s: bubble height (Figure 6a), thickness (Figure 6b), meridian extension (Figure 6c), and circumferential extension (Figure 6d).

### 5.2. PMSQ–HDPE Viscoelastic Behavior Identification Conform to DMA Testing

The least squares method was used to minimize the discrepancies between the experimental and theoretical values during the identification of the relaxation spectrum for PMSQ–HDPE material. This method is described by reducing the objective function defined by Equation (7) where N is the number of experimental data points:(7)$$Z=\overset{N}{\underset{i=1}{\sum }}{\left[\frac{G_{i,exp}^{\prime }-G_{i,th}^{\prime }}{G_{i,exp}^{\prime }}\right]}^{2}+{\left[\frac{G_{i,exp}^{\prime \prime }-G_{i,th}^{\prime \prime }}{G_{i,exp}^{\prime \prime }}\right]}^{2}$$

The parameters $G_{i,\;exp}^{\prime }$ and $G_{i,\;exp}^{\prime \prime }$ represent the dynamic moduli from the experimental data while $G_{i,\;th}^{\prime }$ and $G_{i,th}^{\prime \prime }$ represent the theoretical values given by Christensen’s model (Equation (1)).
(8)$$G_{th}{}^{\prime }\left(\omega \right)=C_{0}+\overset{N}{\underset{i=1}{\sum }}\frac{C_{i}\tau_{i}^{2}\omega^{2}}{2\left(1+\tau_{i}^{2}\omega^{2}\right)}\;\mathrm{and}\;G_{th}{}^{\prime \prime }\left(\omega \right)=\overset{N}{\underset{i=1}{\sum }}\frac{C_{i}\tau_{i}\omega }{2\left(1+\tau_{i}^{2}\omega^{2}\right)}$$

The parameter *C_i_* is the stiffness constant and *τ_i_* the relaxation time associated with the mode *i*, while ω is the frequency. The results obtained are given in Table 3. Figure 7 shows the results of the optimization in comparison with those of the experiment.

## 6. Reliability of Tests on the Viscoelastic Behavior of the PMSQ–HDPE on Thermoforming

In order to identify the most reliable experimental test to characterize the viscoelastic behavior of PMSQ–HDPE for thermoforming applications, we considered the problem of numerical modeling of the thermoforming of a PMSQ–HDPE membrane. For this, we used a circular membrane, similar to the one used in the experiment (see Section 3.1), and a conical mold (see Figure 8). The nonlinear mechanical properties identified for Christensen’s model, with both DMA and biaxial inflation approaches, was used.

### 6.1. Finite Element Analysis

For the analysis, the explicit dynamic finite element method with discretization in space and time was used to simulate the thermoforming of the PMSQ–HDPE membrane. The principle of virtual work was expressed on the undeformed configuration for the inertial effects and internal work.

The spatial and temporal discretizations were both necessary for the virtual work due to the presence of the force of inertia. In the case of spatial discretization, the finite element method approach was considered [21]. However, for temporal discretization, the centered finite difference method, which is conditionally stable, was used. Consequently, the system of equations governing the blowing problem is given by [21]:(9)$$M\;\ddot{u}\left(t\right)=F_{\mathrm{ext}}+F_{\mathrm{grav}}-F_{\mathrm{int}}$$
where
$F_{\mathrm{ext}}$: Global nodal external force vectors$F_{\mathrm{grav}}:$ Global nodal body force vectors$F_{\mathrm{int}}:$ Global nodal internal force vectors**M**: Global mass matrix

The mass matrix **M** can be reduced to a diagonal matrix, **M**d, by using the diagonalization method. For the temporal scheme, we used the finite difference method centered. In this case, Equation (9) can be rewritten as Equation (10):(10)$$u_{i}\left(t+\Delta t\right)=\frac{\Delta t^{2}}{M_{\mathrm{ii}}^{d}}\left(F_{i}^{\mathrm{ext}}\left(t\right)+F_{i}^{\mathrm{grav}}\left(t\right)-F_{i}^{\mathrm{int}}\left(t\right)\right)+2u_{i}-u_{i}\left(t-\Delta t\right)$$

For the stability criterion of system (10), we used the Courant–Friedrichs–Lewy criterion [22].

### 6.2. Plane Stress Assumption and Constitutive Equation

For this, the hypothesis of plane stress and incompressibility of the thermoplastic material was considered. The behavior model used in the simulation was that of Christensen (see Section 3).

### 6.3. Pressure Loading and Van der Waals Equation of State

For the blower modeling of the PMSQ–HDPE membrane, we considered an air flow load. For this purpose, the Redlich–Kwong gas equation of state was considered [23]:(11)$$P\left(t\right)=\frac{n\left(t\right)RT_{g}}{V\left(t\right)-b\;n\left(t\right)}-\frac{n^{2}\left(t\right)a}{V\left(t\right)\left[V\left(t\right)+b\;n\left(t\right)\right]\sqrt{T_{g}}}$$
where:*n*(*t*): the number of gas moles introduced to inflate the thermoplastic-based composite membrane*P*(*t*): the internal pressure*V*(*t*): the volume occupied by the membrane at time t,*T_g_*: the absolute gas temperature*R:* the universal gas constant (=8.3145 J mol^−1^ K^−1^)*a* and *b*: constants evaluated from the critical state of the gas [23]:
(12)$$a=0.42748\;\frac{\bar{R^{2}}T_{c}^{2.5}}{P_{c}}\;and\;b=0.08664\;\frac{\bar{R^{2}}T_{c}}{P_{c}}\;$$
where *Tc* and *P_c_* are the critical temperature and pressure of the gas, respectively. In this study, the assumptions used for the calculation of the dynamic pressure are:(i)Gas temperature is assumed constant (*T_g_*);(ii)The biocomposite sheet temperature is assumed constant (*T_sheet_* = *T_g_*);(iii)At every moment, the pressure between the sheet and the mold is assumed constant (ΔP);(iv)The contact between the biocomposite sheet and the mold is assumed to be a sticky contact as the polymer cools and stiffens rapidly during the sheet/mold contact.

For a reference state in volume (*V*_0_) and number of molds (*n*_0_), Equation (11) becomes:(13)$$P_{0}=\frac{n_{0}RT_{g}}{V_{0}-bn_{0}}-\frac{n_{0}^{2}a}{V_{0}\left[V_{0}+bn_{0}\right]\sqrt{T_{g}}}$$

The dynamic pressure, responsible for inflating the thermoplastic membrane, represents the difference between the internal pressure, induced by the introduction of n(t) mole of gas, and the initial pressure *P*_0_:(14)$$\Delta P\left(t\right)=P\left(t\right)-P_{0}$$

Equation (15) describes, over time, the internal pressure induced by the fluidic charge (air). This pressure, in turn, is responsible for the inflation of the membrane (work). It follows that the following relation expresses the virtual external work in terms of closed volume [24]:(15)$${\delta W}^{\mathrm{exr}}=\Delta P\left(t\right)\;\delta V$$

### 6.4. Analysis of Reliability of Experimental Tests Characterization on Thermoforming

For the study, we considered the thermoforming of a circular PMSQ–HDPE membrane, similar to the one used in experiments for free blowing (with a radius of 4 cm and a thickness of 1.5 mm). For the applied load, we considered a non-linear airflow as shown in Figure 9. The geometries of the mold and the composite sheet discretized by triangular membrane elements are shown in Figure 8. The material temperature was assumed constant at 130 °C. The rheological parameters of the Christensen behavior law are given in Table 2 (relating to the DMA test) and Table 1 (relating to the free biaxial test).

In the following sections of the study, the viscoelastic behavior of the PMSQ-HDPE material relative to the biaxial and DMA tests will be referred to as MB and MD respectively. Figure 10 shows the evolution of the pressure, generated by the airflow, with the volumes for MB and MD. Figure 11 shows the evolution, over time, of its volumes. According to this figure, and contrary to MB, we can see that MD resisted inflation and was unable to ensure its shaping by thermoforming for the treated example, which involved large deformations. To clarify this situation, we have presented in Figure 12a comparison between the two models MB and MD with respect to the principal extensions λ_3_ and the von Mises stress at 0.0143, 0.0293, 0.0443, and 0.0593 s. It can be seen that the action of the air flow on the MD membrane induced, on the one hand, much higher von Mises stresses than those on MB and, on the other hand, a lower stretch. In Table 4, the critical values of the von Mises stresses as well as the principal stretching for the MB and MD models have been provided. Therefore, the MD material is not a candidate for thermoforming and blowing thin, hollow parts that typically induce large deformations. To illustrate this behavior for the MD model, we have presented views of the von Mises constraints at 0.0143, 0.0293, and 0.0443 s in Figure 13, and in Figure 14, we have presented a view of the distribution of its constraints in the proximity of the critical time of 0.0593 s.

It should be pointed out that after the critical time of 0.06 s, the pressure loading had no significant effect on the deformation of the MD membrane, but it had a considerable effect on the stresses. To this effect, we have presented in Figure 15, within the mold, a view of the final shape of the membrane (including the contact nodes), and a view of the von Mises stresses. The MD became quasi-rigid.

For the rest of the study, only the MB material is considered. Figure 16 illustrates the evolution of the nodes, which were in contact with the mold during the forming process, represented by the black dots.

When simulating the thermoforming of a thin part, it is important to predict the thickness and stress distributions in the molded part. In fact, the predictions of the residual stresses and dimensional stability of the final shape of the molded part are closely related to the estimated stresses. In addition, the effect of localized thinning of the deformed membrane is usually accompanied by an increase in the Cauchy stresses (or actual stresses). For this purpose, we have presented in Figure 17 the von Mises stress distribution and the main extensions on the trace of the thermoformed part. The maximum value of the von Mises constraint is of the order 5.4 MPa and is located at the positions 2.5 and 4.5 m. For this critical stress value, the principal stretch λ_3_ is 0.093. In Figure 18, we have presented an overview of the von Mises stresses (Figure 18a) and the main extensions (Figure 18b) induced in the thermoformed part.

In light of the results presented above, the following remarks can be made about the use of experimental tests for viscoelastic identification relative to Christensen’s model:-The experimental test used for the construction of the constitutive behavior law of polymers plays a key role on the qualities of the results;-The results obtained by the mechanical blowing test, which induces deformation modes similar to those encountered in thermoforming, seem to be the most appropriate;-The construction of viscoelastic laws from DMA is more suitable for small deformations for thermoforming applications;-The choice to use the finite element method with a pressure load, which is derived from a thermodynamic law, is judicious for the integrated analysis in large deformations of the forming of a thin part;-Experimental temperature can improve the quality of viscoelastic identification for thermoforming applications. The material becomes softer.

## 7. Conclusions

The study was conducted on the reliability of the experimental method for viscoelastic identification of a nanocomposite reinforced with Polymethylsilsesquioxane nanoparticles (PMSQ–HDPE). To do so, two tests of different nature were used. One was based on free inflation of the membrane and the other on a dynamic mechanical test (DMA). The experiments were carried out at a temperature of 130 °C. The material constants for Christensen’s model were determined by the least squares optimization. The comparative study of viscoelastic behavior of PMSQ–HDPE shows that the biaxial test is more appropriate for the construction of a behavior law for applications in thermoforming. Concerning the viscoelastic identification obtained from the rheological data of the DMA, it does not seem to be able to represent the thermoforming of a part which requires large deformations.

Following this study, comparative studies between the DMA and the free blowing should be carried out at temperatures above 130 °C for viscoelastic identification. This will make it possible to characterize the effect of temperature on the reliability of the tests in thermoforming.

## Figures and Tables

**Figure 1 polymers-12-02753-f001:**
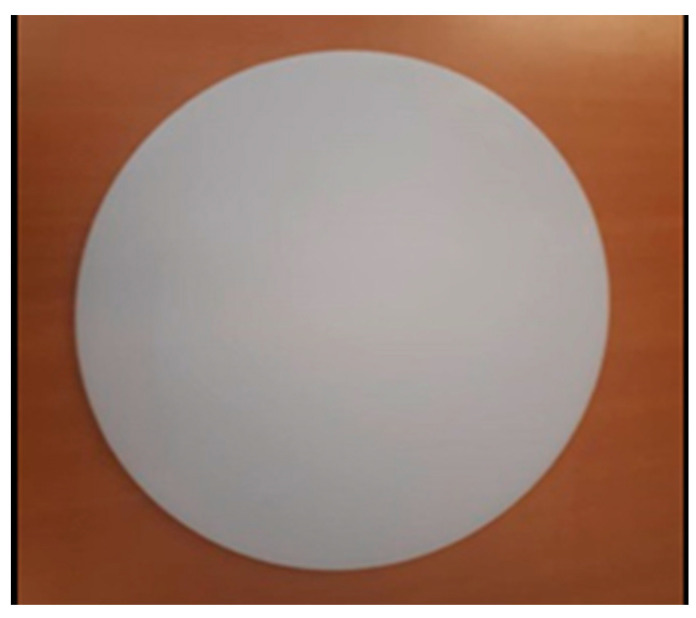
PMSQ–HDPE membrane.

**Figure 2 polymers-12-02753-f002:**
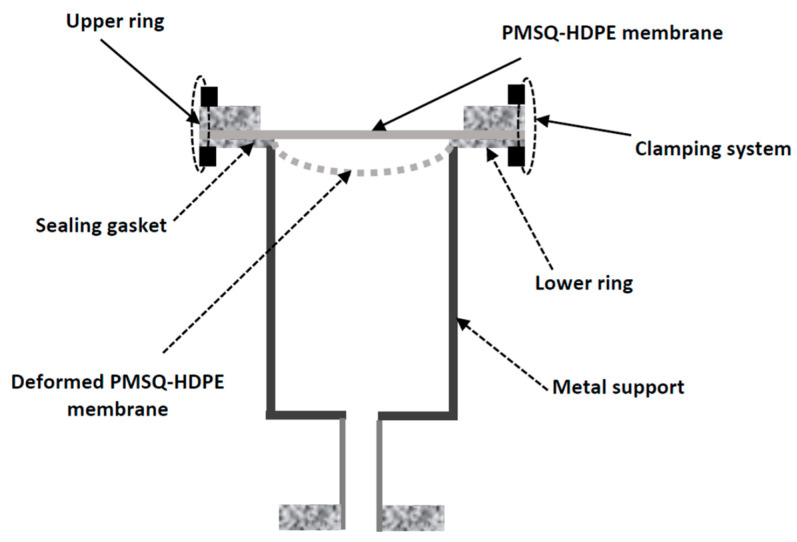
Fixation module.

**Figure 3 polymers-12-02753-f003:**
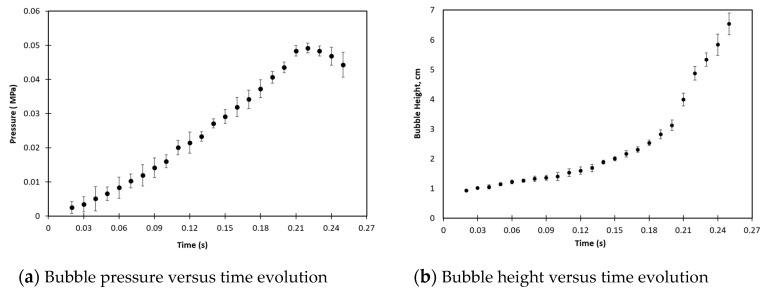
Bubble pressure and bubble height versus time evolution.

**Figure 4 polymers-12-02753-f004:**
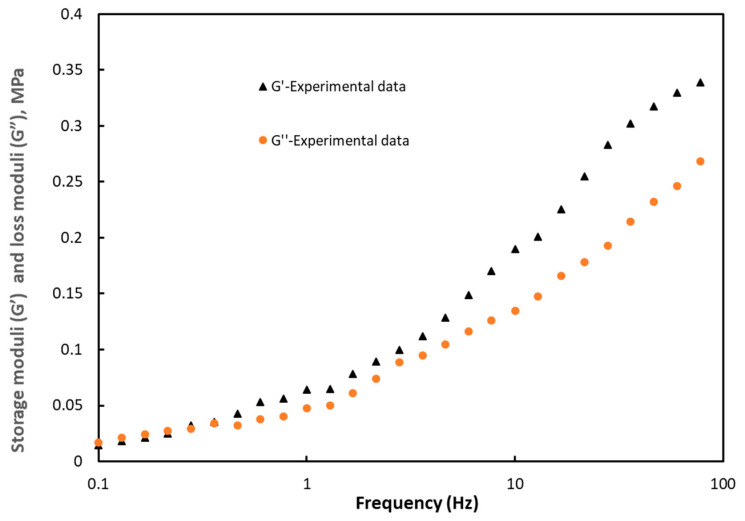
Experimental results of the storage moduli (G’) and loss moduli (G”) as a function of the frequency.

**Figure 5 polymers-12-02753-f005:**
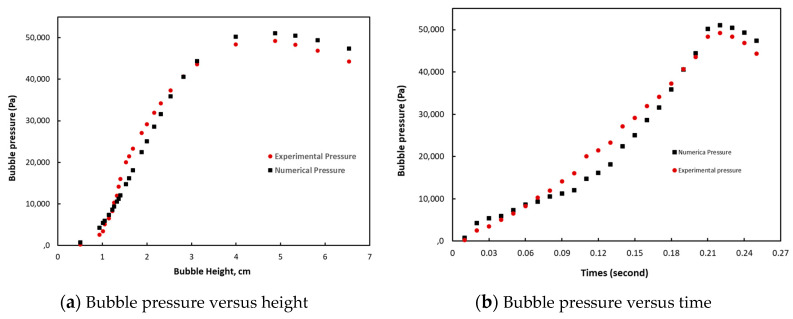
Results of the optimization with the experimental data: (**a**) bubble pressure vs bubble height and (**b**) bubble pressure vs times.

**Figure 6 polymers-12-02753-f006:**
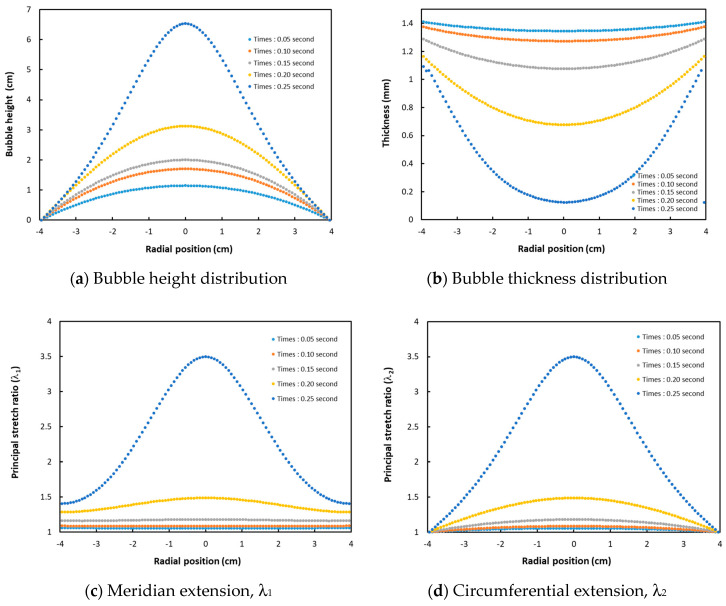
Geometrical PMSQ–HDPE membrane according to Christensen’s model at 0.05, 0.10, 0.15, 0.20, and 0.25 s: (**a**) Bubble height, (**b**) Bubble thickness, (**c**) Stretch ration λ_1_ and (**d**) Stretch ration λ_2_.

**Figure 7 polymers-12-02753-f007:**
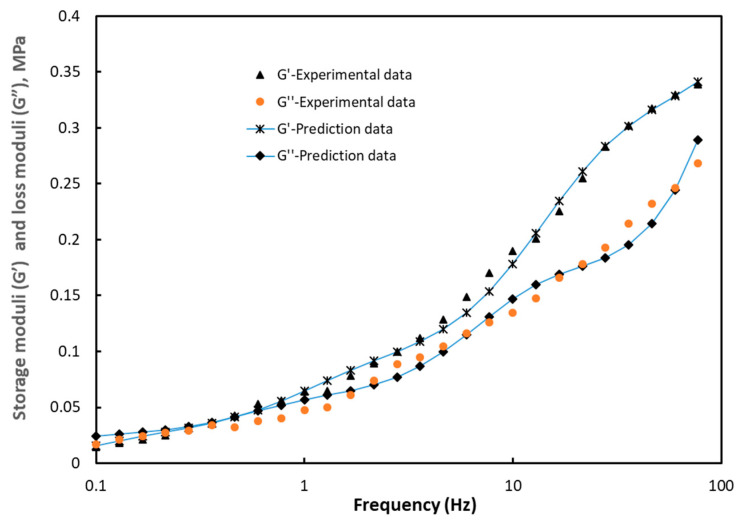
Results of the optimization with the experimental storage moduli G’ and loss moduli G”.

**Figure 8 polymers-12-02753-f008:**
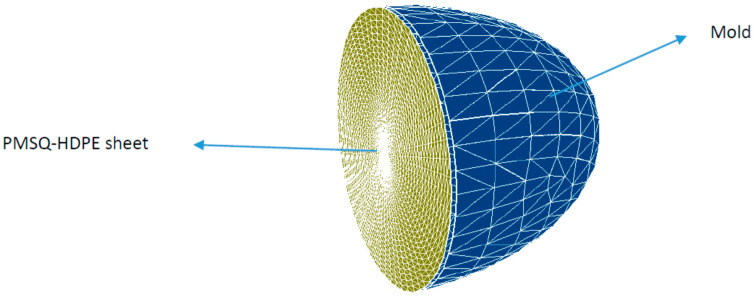
Geometries of the mold and the wood plastic composite (WPC) sheet.

**Figure 9 polymers-12-02753-f009:**
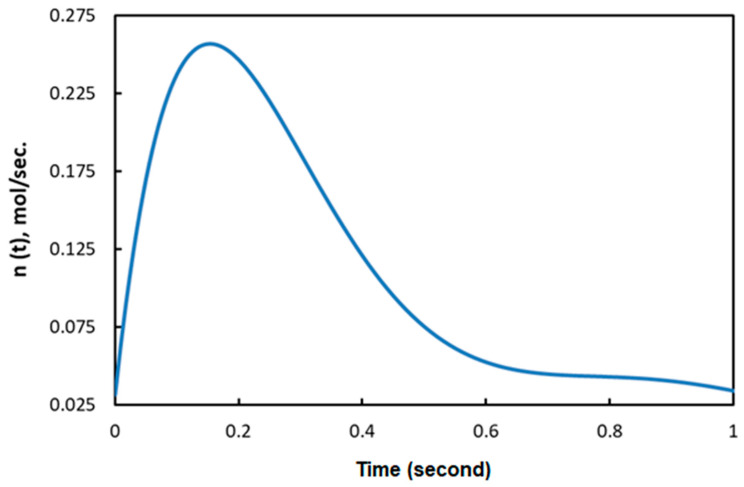
Airflow versus time.

**Figure 10 polymers-12-02753-f010:**
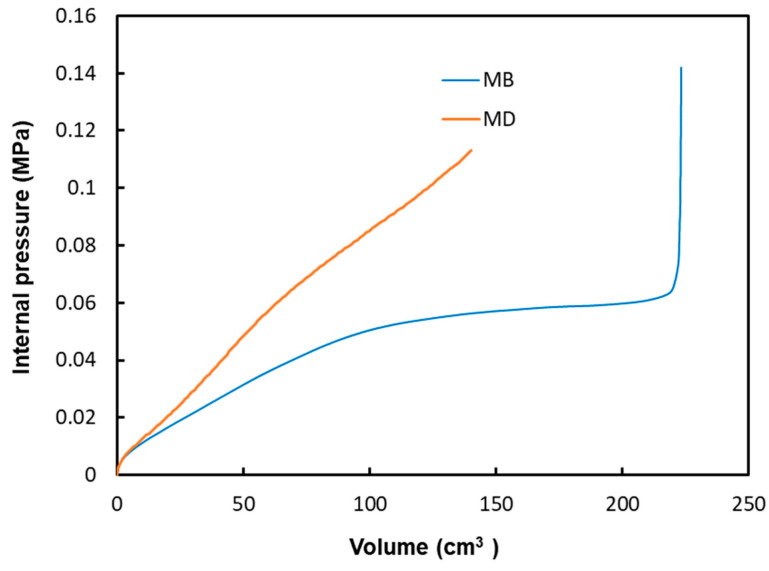
Internal pressure versus volume of PMSQ–HDPE.

**Figure 11 polymers-12-02753-f011:**
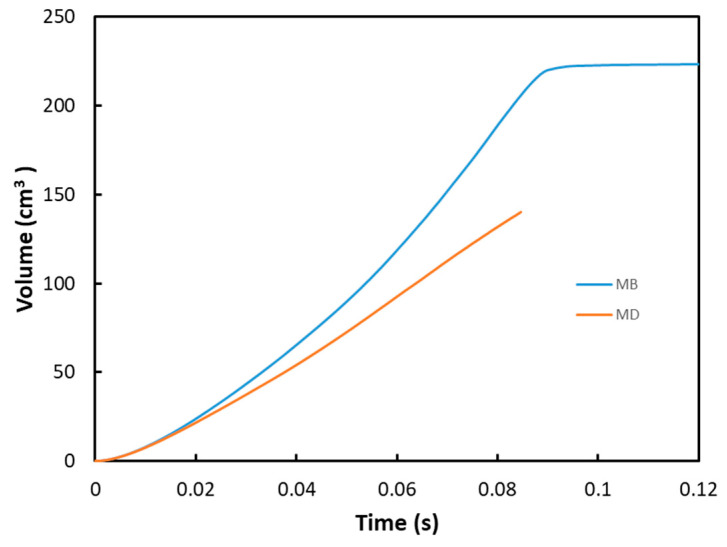
Volume evolution with time of PMSQ–HDPE.

**Figure 12 polymers-12-02753-f012:**
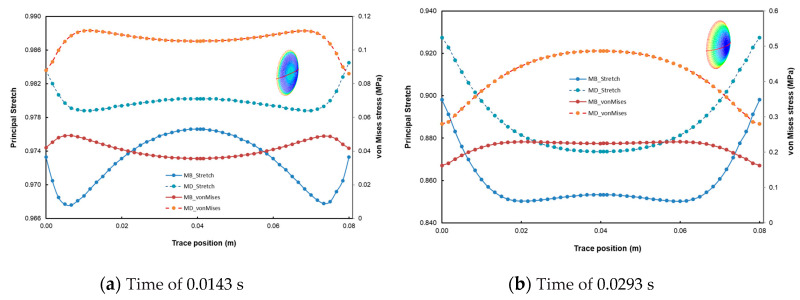
Transient evolution of the principal stretch and von Mises stresses at different instants: (**a**) time = 0.0143 s, (**b**) time = 0.0291 s, (**c**) time = 0.0443 s and (**d**) time = 0.0593 s.

**Figure 13 polymers-12-02753-f013:**
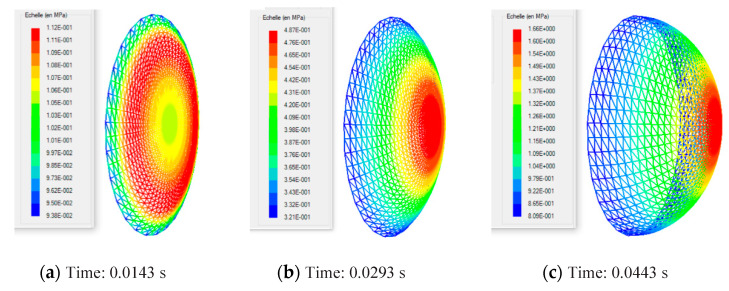
MD model: von Mises stresses induced in PMSQ–HDPE at times of 0.0143 (**a**), 0.0293 (**b**), and 0.0443 s (**c**).

**Figure 14 polymers-12-02753-f014:**
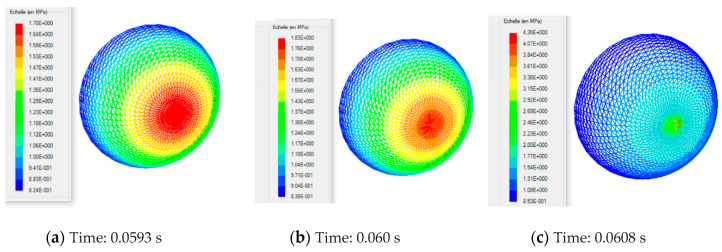
Von Mises constraints induced in the HDPE–PMSQ with the MD model in the time-critical neighborhood 0.0593 s.

**Figure 15 polymers-12-02753-f015:**
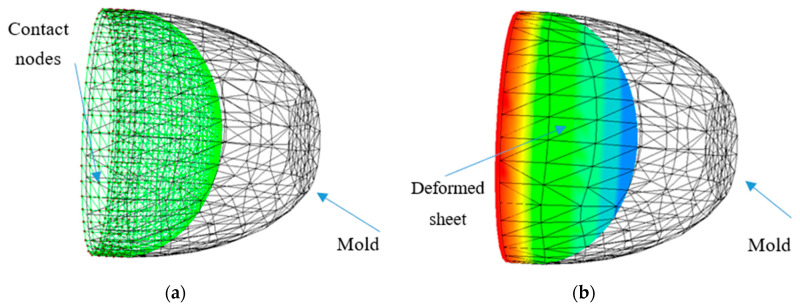
Shape of the deformed MD and principal stretch λ_3_ in the time-critical neighborhood 0.06 s. (**a**) Final shape of the deformed MD. (**b**) Distribution of the principal extensions λ_3_ in the final deformed MD.

**Figure 16 polymers-12-02753-f016:**
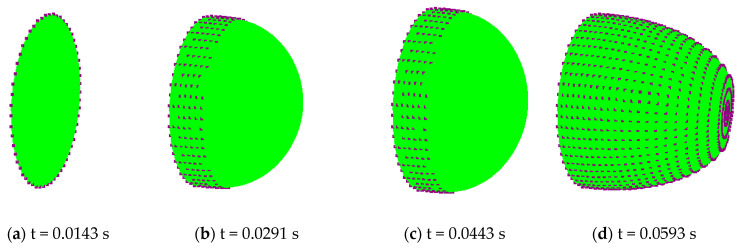
Evolution of the distribution of the contact nodes with the mold at at different instants: (**a**) time = 0.0143 s, (**b**) time = 0.0291 s, (**c**) time = 0.0443 s and (**d**) time = 0.0593 s.

**Figure 17 polymers-12-02753-f017:**
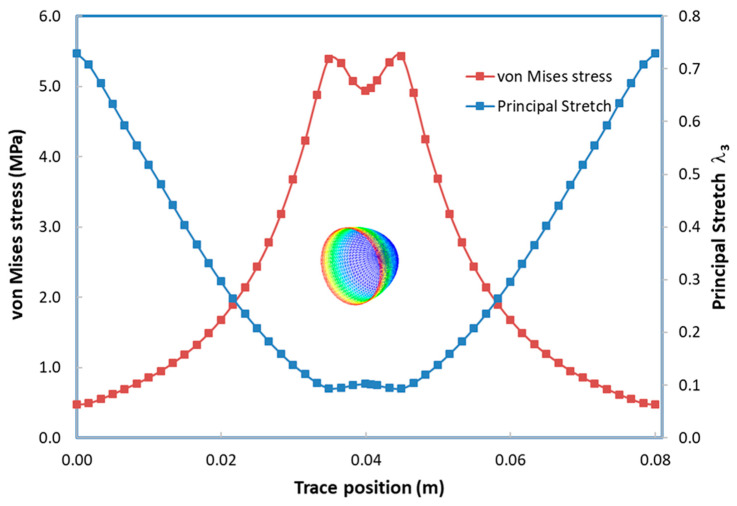
Von Mises stress and stretch ratio, λ_3_, on the half plane of symmetry at the end of the forming cycle.

**Figure 18 polymers-12-02753-f018:**
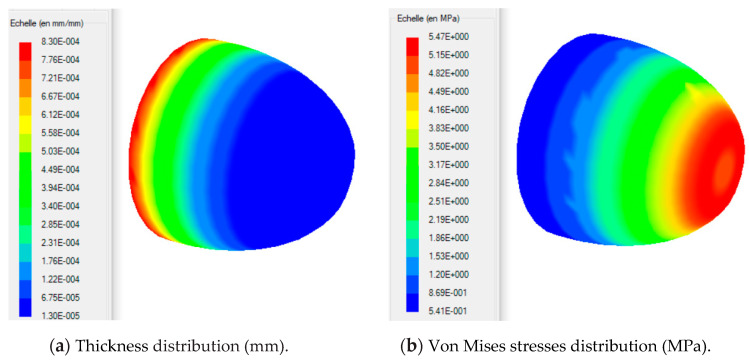
Thickness and von Mises stresses distributions in the thermoformed part: (**a**) Thickness distribution and (**b**) Von Mises stresses distribution.

**Table 1 polymers-12-02753-t001:** Tensile mechanical properties of high-density polyethylene–polymethylsilsesquioxane (HDPE–PMSQ) nanocomposites [19].

% PMSQ–HDPE	Elastic Modulus (MPa)	Yield Stress (MPa)	Elongation at Break (%)
0.0%	1031 ± 26	26.8 ± 0.2	39.2 ± 2.3
0.5%	1064 ± 60	27.9 ± 0.3	47.2 ± 3.1
1.0%	1115 ± 54	30.1 ± 0.1	41.1 ± 2.3

**Table 2 polymers-12-02753-t002:** Materials constants for PMSQ–HDPE at 130 °C.

C_0_ (MPa)	C_1_ (MPa)	τ_1_ (s)
0.71694	0.00001	772.00037

**Table 3 polymers-12-02753-t003:** Stiffness modulus and relaxation time for the PMSQ–HDPE nanocomposite at T = 130 °C.

PMSQ–HDPE at 130 °C					
C_0_ (MPa)	C_1_ (MPa)	C_2_ (MPa)	C_3_ (MPa)	C_4_ (MPa)	C_5_ (MPa)
−0.000633	6.414907	0.294631	0.170894	0.132937	0.062403
	τ_1_(s)	τ_2_(s)	τ_3_(s)	τ_4_(s)	τ_5_(s)
	0.01	0.06	0.1	1.0	10.0

**Table 4 polymers-12-02753-t004:** Critical values of von Mises stress and principal stretch λ_3__._

Time (s)	Von Mises Stress MPa		Principal Stretch λ_3_	
	MB Model	MD Model	MB Model	MD Model
0.0143	0.04923	0.1116	0.9676	0.9788
0.0293	0.2298	0.4866	0.8503	0.8736
0.0443	0.6185	0.9467	0.6222	0.6966
0.0593	1.158	1.658	0.408	0.5307

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
