# Peer review of "Reliability of Free Inflation and Dynamic Mechanics Tests on the Prediction of the Behavior of the Polymethylsilsesquioxane–High-Density Polyethylene Nanocomposite for Thermoforming Applications"

_polymers, 2020, doi:10.3390/polym12112753_

Round 1
Reviewer 1 Report
Thank you for submitting your work. I have made a thorough review and is happy with the technicality of the paper and the explanations. However, I have some questions which are required to carefully answered and corrected in the manuscript.
- Please improve the English and grammar throughout the manuscript.
- Abstract should be rewritten. it does not convey the entire work. Try to be little qualitative as well.
- At the end of the introduction section, novelty of the work should be clearly presented.
- Add a table summarising the material properties. It is not sufficient to cite from the previous work.
- How it was ensured that the temperature at the surface of the membrane was uniform (Page 3 Line 110)
- Why shear experiment was preferred for DMA?
- Page 10, Line 148, assumptions based on? citation? references?
- Page 11 (Line 273-274), not very clear.
- Figure 18, thickness, check spelling
- Conclusion section is very weak and should be written in more detail.
Author Response
Reviewer 1: Thank you for submitting your work. I have made a thorough review and is happy with the technicality of the paper and the explanations. However, I have some questions which are required to carefully answered and corrected in the manuscript
Point 1: Please improve the English and grammar throughout the manuscript.
Answer 1: Thanks for the remark. The revision has been made (see the updated version of the manuscript).
Point 2: Abstract should be rewritten. it does not convey the entire work. Try to be little qualitative as well.
Answer 2: The summary has been updated. Here is the new version of the summary:
Abstract:Numerical modelling of the thermoforming process of polymeric sheets requires precise knowledge of the viscoelastic behavior under the conjugate effect applied pressure and temperature. Using two different experiments, bubble inflation and dynamic mechanical testing on an HDPE nanocomposite reinforced with Polymethylsilsesquioxane nanoparticles (PMSQ-HDPE), material constants for the Christensen model are determined by the least squares optimization. The viscoelastic identification relative to the inflation test seems to be the most appropriate for the numerical study of thermoforming of a thin PMSQ-HDPE part. For this purpose, the finite element method is considered
Point 3: At the end of the introduction section, novelty of the work should be clearly presented
Answer 3: The section at the end of the introduction is updated. Here is the added section:
The deformations induced in thermoplastics, in thermoforming process, are significant and, in general, of a biaxial nature. However, several works encountered in the literature on the construction of viscoelastic constitutive laws are based on experimental data from DMA. Thus, the following question arises: are the rheological data resulting from DMA tests reliable for the construction of a viscoelastic law? It is in this context that the present work is oriented and aims at a study on the reliability of the results obtained from two experimental tests: one is based on the inflation of the membrane and the other on a dynamic mechanical test (DMA). The two experimental tests are carried out at a temperature of 130°C. For the viscoelastic characterization, we considered the Christensen model [5]. The mechanical parameters were identified using the Levenberg-Marquardt algorithm [17].
For the comparative study of the reliability of the results of the viscoelastic identification, compared to each experimental test, we considered the numerical modeling of the thermoforming of a thin part in PMSQ-HDPE. For this purpose, the finite element method is considered.
Point 4: Add a table summarising the material properties. It is not sufficient to cite from the previous work.
Answer 4: In the updated version of the manuscript, a table has been added containing the mechanical properties of PMSQ-HDPE:
Table 1: Tensile mechanical properties of HDPE-PMSQ nanocomposites [19]
|
% PMSQ-HDPE |
Elastic modulus (MPa) |
Yield stress (MPa) |
Elongation at beak (%) |
|
|
0.0% |
1031±26 |
26.8±0.2 |
39.2±2.3 |
|
|
0.5% |
1064±60 |
27.9±0.3 |
47.2±3.1 |
|
|
1.0% |
1115±54 |
30.1±0.1 |
41.1±2.3 |
|
Point 5: How it was ensured that the temperature at the surface of the membrane was uniform (Page 3 Line 110).
Answer 5: To make sure of the surface temperature, we used an infrared camera.
Point 6: Why shear experiment was preferred for DMA
Answer 6: We often use shear (G', G''), tensile (E' and E'') or bending tests for DMA. However, for Christensen's viscoelastic model, the parameters E' and E'' are respectively proportional to G' and G''. Therefore, the results obtained in shear (G', G'') or in tension (E', E'') will induce the same mechanical parameters for Christensen's model.
Point 7: Page 10, Line 148, assumptions based on? citation? references?
Answer 7. We have used the same assumptions as in reference 20.
Point 8: Page 11 (Line 273-274), not very clear..
Answer 8: Before forming the polymeric sheet, the latter is deposited on a mold. Only the edges of the sheet are then in contact with the mold. The material in contact with the mold does not move during forming (boundary condition for simulation). However, we have decided, to eliminate inappropriate interpretations, to delete lines 273 and 274 in the manuscript
Point 9: Figure 18, thickness, check spelling
Answer 9: The correction has been made
Point 10: Conclusion section is very weak and should be written in more detail
Answer 10: Here is the new version:
The study is conducted on the reliability of the experimental method for viscoelastic identification of a nanocomposite reinforced with Polymethylsilsesquioxane nanoparticles (PMSQ-HDPE). To do so, two tests of different nature are used: one is based on free inflation of the membrane and the other on a dynamic mechanical test (DMA). The experiments were carried out at a temperature of 130 oC. The material constants for the Christensen model are determined by the least Squares Optimization. The comparative of viscoelastic behavior of PMSQ-HDPE shows that the biaxial test is more appropriate for the construction of a behavior law for applications in thermoforming. Concerning the viscoelastic identification obtained from the rheological data of the DMA test, it does not seem to be able to represent the thermoforming of a part which requires large deformations.
Following this study, it will be desirable that comparative studies, between the DMA test and the free blowing, be carried out at temperatures above 130 °C for viscoelastic identification. This will make it possible to characterize the effect of temperature on the reliability of the tests in thermoforming.
Thank you for the very constructive comment
Pr Erchiqui
Reviewer 2 Report
This manuscript reports on the reliability of experimental methods on the characterization of the viscoelastic behaviour of a polymer nanocomposite. For the study, the results of two different experimental methods, in particular the study of the inflation of the membrane and a dynamic mechanical test, are compared to numerical results using the finite element method.
The study is of interest but it needs to be improved. Authors should be careful with the presentation because there are too many mistakes and typos which at some point make difficult to follow the reading. Also, there are too many figures (19) and some of them could be shown as supplementary instead of being in the main document. Also, some of them could be merged in one, for instance this is the case of figures 3 and 4, which should be plotted with the same time scale or even in the same figure.
Besides the large amount of figures, in many of them there are errors. Thus:
- Correct labels and figure caption of figure 5
- Correct figure 6 (Figure 5b is also labelled as figure 5a)
- Correct figure 8
- There are two figures numbered as 17
In general, homogenize format of figures and figure captions
Additionally, correct titles and subtitles (for instance after 6.3, 6.5 should appear instead of 6.2)
Author Response
Polymers
Revision of manuscript: polymers-990835
“Reliability of free inflation and dynamic mechanics tests on the prediction of the behavior of the PMSQ-HDPE nanocomposite for thermoforming applications”
REVIEWERS' REMARKS
Reviewer 2: The study is of interest but it needs to be improved. Authors should be careful with the presentation because there are too many mistakes and typos which at some point make difficult to follow the reading. Also, there are too many figures (19) and some of them could be shown as supplementary instead of being in the main document. Also, some of them could be merged in one, for instance this is the case of figures 3 and 4, which should be plotted with the same time scale or even in the same figure.
Besides the large amount of figures, in many of them there are errors. Thus:
- Correct labels and figure caption of figure 5
- Correct figure 6 (Figure 5b is also labelled as figure 5a)
- Correct figure 8
- There are two figures numbered as 17
In general, homogenize format of figures and figure captions
Additionally, correct titles and subtitles (for instance after 6.3, 6.5 should appear instead of 6.2)
Answer: Thanks for the remark. The revision has been made (see the updated version of the manuscript).
Figures 3 and 4 have been combined into a single figure 3. The remarks below have been made:
- Corrected the labels and legend of figure 5
- Correction of the labels of figure 6
- Correction of figure 8
- Update of the numbering of 17 and 18
- Homogenization of figures
Thank you for the very constructive comment
Pr Erchiqui
Round 2
Reviewer 2 Report
In the revised version of the manuscript authors claimed that they have implemented several changes, but it does not seem to be the case. In particular:
- In the previous revision it was suggested that figures 3 and 4 (figure 3a and 3b in the present version) should be shown using the same time scale, so please modify them.
- It was also mentioned that label in figure 5 (now figure 4) should be corrected but they are the same: G’’ twice instead of G’ and G’’. Also the caption (although it has been modified it is incorrect: storage moduli (G’) instead of (G’))
- Caption of figure 5 has not been modified and Figure b is labelled as figure a
- Labels of figure 7 are still incorrect
Author Response
Thank you for your comments.
Corrections have been made in the updated version
- Same time scale for figures 3a and 3b d
- The label in figure 4 (has been corrected as well as the legend).
- The legend of figure 5 has been updated as well as the labels (figures a and b)
- The labels in Figure 7 have been corrected.
Round 3
Reviewer 2 Report
Manuscript has been adequately revised and corrections have been implemented.